# Effect of Early Rehabilitation on Physical Function in Patients Undergoing Coronary Artery Bypass Grafting: A Nationwide Inpatient Database Study

**DOI:** 10.3390/jcm10040618

**Published:** 2021-02-06

**Authors:** Hiroyuki Ohbe, Kensuke Nakamura, Kazuaki Uda, Hiroki Matsui, Hideo Yasunaga

**Affiliations:** 1Department of Clinical Epidemiology and Health Economics, School of Public Health, The University of Tokyo, 7-3-1 Hongo, Bunkyo-ku, Tokyo 1130033, Japan; hohbey@gmail.com (H.O.); udakazuaki-tky@umin.ac.jp (K.U.); ptmatsui-tky@umin.ac.jp (H.M.); yasunagah-tky@umin.ac.jp (H.Y.); 2Department of Emergency and Critical Care Medicine, Hitachi General Hospital, 2-1-1, Jonantyo, Hitachi, Ibaraki 3170077, Japan

**Keywords:** rehabilitation, physical impairments, coronary artery bypass grafting, intensive care unit, post-intensive care syndrome, observational study

## Abstract

It is unclear when to begin rehabilitation after coronary artery bypass grafting (CABG) in the intensive care unit (ICU). Using the Japanese Diagnosis Procedure Combination inpatient database from 2010 to 2018, we identified adult patients who underwent a CABG and who were admitted to the ICU for ≥3 consecutive days from the date of their CABG. Patients who started any rehabilitation program prescribed by physicians or therapists within 3 days of CABG were defined as the early rehabilitation group, and the remaining patients were defined as the usual care group. We identified 30,568 eligible patients, with 13,150 (43%) patients in the early rehabilitation group. An inverse probability of treatment weighting analyses showed that the Barthel Index score at discharge in the early rehabilitation group was significantly higher than that in the usual care group (difference: 3.2; 95% confidence interval: 1.5–4.8). The early rehabilitation group had significantly lower in-hospital mortality, total hospitalization costs, length of ICU stay, and hospital stay vs. the usual care group. Our results suggested that early rehabilitation by physicians or therapists beginning within 3 days of CABG was safe, as suggested by the low mortality and improved physical function in patients who underwent CABG.

## 1. Introduction

Patients admitted to intensive care units (ICUs) can experience physical disabilities, cognitive impairments, or psychological difficulties, which are related to post-intensive care syndrome (PICS). PICS occurs within 24 h of ICU admission and progresses [1,2,3], and is induced by critical illness, treatments, ongoing health conditions, and a lack of movement during the ICU stay [4]. Patients with PICS have a very poor long-term health-related quality of life; therefore, maintaining activities of daily living (ADLs) during hospitalization is important [5]. Early rehabilitation, mobilization, or active exercise programs are increasingly being provided to maintain ADLs and independence for critically ill patients in ICUs [6].

Coronary artery bypass grafting (CABG) is used for multivessel diseases that are difficult to revascularize using a percutaneous coronary intervention. Following CABG, patients are more likely to have physical impairments and PICS because of their low cardiac function, the risk of ischemia, and the surgical invasiveness [7]. Therefore, cardiac rehabilitation has been widely performed with patients after a CABG. The timing of cardiac rehabilitation is divided into phase 1 (acute stage: within 7 days of onset), phase 2 (healing stage: from 1 week to 6 months after onset), and phase 3 (healed stage: >6 months after onset) [8]. Previous studies showed that cardiac rehabilitation demonstrated beneficial effects on physical function [9,10], cardiopulmonary function [11,12], cardiopulmonary symptoms [13], stenoses in coronary arteries [14], inflammatory response [15], and autonomic function [16]. However, after CABG, the beneficial effects of cardiac rehabilitation were demonstrated mainly in phases 2 and 3 [7]. Evidence supporting a beneficial effect of phase 1 cardiac rehabilitation after CABG is limited, and it is not well known when to start phase 1 rehabilitation after CABG. Previous small, randomized studies showed conflicting results of the effects of early rehabilitation within 3 days after CABG [17,18,19,20,21]. Rehabilitation too soon after CABG may increase the risk of coronary ischemia or inhibit wound healing [22]. Therefore, this study aimed to assess whether early rehabilitation beginning within 3 days of CABG improved physical function at discharge by using a large national inpatient database in Japan.

## 2. Materials and Methods

This was an observational study using an administrative database, where the study conformed to the REporting of studies Conducted using Observational Routinely-collected Data (RECORD) statement reporting guidelines [23]. This study was conducted in accordance with the amended Declaration of Helsinki and was approved by the Institutional Review Board of the University of Tokyo (approval number: 3501-(3); 25 December 2017). Because the data were anonymous, the Institutional Review Board waived the requirement for informed consent. No information about individual patients, hospitals, or treating physicians was available.

We used the Japanese Diagnosis Procedure Combination inpatient database, which contains administrative claims data and discharge abstracts from more than 1200 acute care hospitals and covers approximately 90% of all tertiary emergency hospitals in Japan [24]. The database includes age, sex, smoking history, body weight, body height, level of consciousness at admission, diagnoses (main diagnosis, comorbidities present at admission, and complications arising after admission) that were recorded according to the International Classification of Diseases Tenth Revision codes, treatments recorded according to Japanese medical procedure codes, discharge status, and activities of daily living scores that could be converted to Barthel Index scores at both admission and discharge [24]. To optimize the accuracy of the recorded diagnoses, attending physicians are required to report the objective evidence for diagnoses for treatment reimbursement. In a previous validation study that evaluated the records of diagnoses and procedures in the database, the specificity of the diagnoses exceeded 96%, while the sensitivity of the diagnoses ranged from 50 to 80%; the specificity and sensitivity of procedures both exceeded 90% [25].

Using the Japanese Diagnosis Procedure Combination inpatient database from 1 July 2010 to 31 March 2018, we identified patients who underwent CABG and who were admitted to the ICU for more than 3 consecutive days from the date of CABG. We excluded patients aged <18 years and those who received cardiopulmonary resuscitation within 3 days of their CABG.

## 3. Group Assignment

Patients beginning any rehabilitation program prescribed by physicians, physical therapists, or occupational therapists within 3 days of their CABG were defined as the early rehabilitation group. The remaining patients were defined as the usual care group. In the usual care group, nurses provided rehabilitation as needed, without a formal rehabilitation program. In Japan, rehabilitation by physicians, physical therapists, or occupational therapists is reimbursed by the universal healthcare insurance system, and one rehabilitation unit is considered to be 20 min [26].

## 4. Outcomes and Covariates

The primary outcome was patients’ physical function at discharge, as measured via the Barthel Index score using continuous scales [27]. Patients who died during hospitalization were assigned scores of 0 for the Barthel Index [6]. The secondary outcomes were in-hospital mortality, length of ICU stay, length of hospital stay, and total hospitalization costs.

The covariates were age, sex, smoking history, body mass index at admission, Barthel Index score at admission, Japan Coma Scale score at admission [28], calendar year, ambulance use, emergency admission, length of stay until surgery, annual hospital volume, teaching hospital, Charlson Comorbidity Index score [29], comorbidities, surgical characteristics, and treatments within 2 days of the CABG.

## 5. Multiple Imputation

Some patients had missing data for body mass index, smoking history, Barthel Index at admission, or Barthel Index at discharge. Therefore, we performed multiple imputation to address missing values that were assumed to be missing at random [30]. We created 20 imputed datasets constituting all the covariates and outcomes using multiple imputation via chained equations for the missing data. We used the “mi impute chained” command in STATA/MP 16.0 software (StataCorp, College Station, TX, USA) [31]. We combined the imputation estimates and standard errors according to Rubin’s rule [32].

## 6. Statistical Analysis

We used the inverse probability of treatment weighting (IPTW) to compare outcomes of the early rehabilitation group with those of the usual care group [33,34]. We applied a multivariable logistic regression model to predict the propensity scores for patients undergoing rehabilitation within 3 days of their CABG using all variables listed in Table 1 as predictor variables. We used the stabilized average treatment effect weight, which allowed us to maintain the total sample size of the original data and provided a more accurate interval estimate of the variance of the main effect and controls for type I error compared with the non-stabilized IPTW [35].

We calculated the absolute standardized differences of each covariate in the unweighted and weighted cohorts to confirm the balance of the distribution of the covariates between the early rehabilitation and usual care groups. When the absolute standardized differences between the two groups were less than 10%, we considered the imbalance in the distribution of the covariates to be negligible [36].

We used a weighted generalized linear model to compare the outcomes, with cluster-robust standard errors and individual treating hospitals as clusters. We calculated the differences and their 95% confidence intervals for outcomes using the identity link function.

We performed two sensitivity analyses to confirm the robustness of our results. First, we performed a complete case analysis that excluded patients with any variable with missing values. Second, we performed propensity-score-adjusted analyses in which we employed a multivariable regression model using primary or secondary outcomes as the dependent variable with early rehabilitation within 3 days of CABG and the estimated propensity scores in the main analyses as the independent variables.

Continuous variables were presented as a mean and standard deviation (SD), while categorical variables were presented as a number and percentage. We considered all reported *p*-values as two-sided and *p* < 0.05 as statistically significant. All analyses were performed using STATA/MP 16.0 software (StataCorp).

## 7. Results

After applying the inclusion and exclusion criteria, 30,568 patients were included in this study (Figure 1). Of these, 13,150 (43%) patients were assigned to the early rehabilitation group (rehabilitation beginning within 3 days of CABG). The frequency of rehabilitation within 7 days of CABG was 7.3 units (SD: 5.1 units) in the early rehabilitation group and 1.3 units (SD: 2.1 units) in the usual care group.

The patients’ characteristics and the numbers of patients with missing values for each variable are presented in Appendix A show the results of comparing the distributions of the variables between patients with complete and incomplete data. Multiple imputation created 20 imputed datasets, where the patients’ characteristics for one of the imputed datasets are shown in Table 1. After the IPTW, the patients’ characteristics were well-balanced between the two groups. The propensity score distribution between the two groups was adequately balanced after the IPTW (Appendix A).

Table 2 shows the outcomes in the unweighted and weighted cohorts. After the IPTW, the Barthel Index scores at discharge in the early rehabilitation group were significantly higher than those in the usual care group (difference: 3.2; 95% confidence interval: 1.5–4.8). The early rehabilitation group had lower in-hospital mortality, total hospitalization costs, length of ICU stay, and hospital stay vs. the usual care group. The results of the sensitivity analyses for the complete case analysis and propensity score adjustment (Table 3 and Appendix A, respectively) were similar to those of the main analysis.

## 8. Discussion

The present study using nationwide real-world data showed that early rehabilitation beginning within 3 days of CABG prescribed by physicians or therapists was significantly associated with improved ADLs at discharge after the CABG. Furthermore, early rehabilitation was significantly associated with lower in-hospital mortality, total hospitalization costs, length of ICU stay, and hospital stay.

Similar to previous studies, our results suggested that early cardiac rehabilitation within 3 days of CABG effectively maintained ADLs in patients after their CABG [17,18,19]. Previous studies also revealed that early rehabilitation was more effective in patients undergoing CABG than for those with other cardiovascular diseases because patients undergoing CABG had several narrowed or blocked coronary vessels, underwent marked surgical stress, were exposed to longer immobilization periods, and required longer recovery periods [19,37]. Although our study did not evaluate the intensity of the rehabilitation, one study showed that even small amounts of exercise improved physical function in patients who underwent CABG [17]. Generally, earlier cardiac rehabilitation programs are more effective in the ICU after CABG surgery.

Our results indicated that cardiac rehabilitation in the early stage in phase 1 after a CABG was safe. Similarly, recent studies showed that early ICU rehabilitation was considered safe without increasing the complications of coronary ischemia or heart failure when the rehabilitation was administered with appropriate timing and discontinuation criteria [38,39,40]. Previous studies also showed that the hemodynamic responses triggered by very early cardiac rehabilitation were similar and safe, regardless of whether the patients had low or high cardiac function after their CABG [41,42]. Therefore, early rehabilitation after CABG can be administered safely, even in patients with low cardiac function, as well as in other patients in the ICU.

Early cardiac rehabilitation was associated with a lower length of ICU stay, hospital stay, and total hospitalization costs in our study. Guidelines for cardiac diseases recommend starting cardiac rehabilitation during phase 1; however, very early rehabilitation in phase 1, such as beginning on day 2 or 3 after CABG, is not recommended because of limited evidence [22]. The real-world data in our study suggested that early rehabilitation within 3 days of CABG was clinically effective and economical. Further studies evaluating the efficacy and safety of early rehabilitation and when to begin are warranted.

There are several limitations to the present study. First, because we used a real-world Japanese database, the decision whether to start rehabilitation was made by individual clinicians according to their own criteria. The decision to start rehabilitation itself may be an indicator of severity, and the reasons for making this decision were not standardized, which led to confounding by indication. We attempted to control for this confounding by indication using propensity score analyses. However, we were unable to control for possible unmeasured variables, such as vital signs or cardiac function. Second, because the database contained information collected only during hospitalization, we were unable to analyze long-term outcomes after discharge. We were also unable to assess the detailed effects and safety of factors that may have contributed to improved mortality, such as cardiopulmonary function, cardiopulmonary symptoms, stenoses in coronary arteries, the inflammatory response, or autonomic function. Third, we could not evaluate the exercise intensity of the rehabilitation.

## 9. Conclusions

This study using a national inpatient database suggested that early rehabilitation beginning within 3 days of CABG was safe, as evidenced by the low mortality and improved physical function at discharge in patients who underwent CABG.

## Figures and Tables

**Figure 1 jcm-10-00618-f001:**
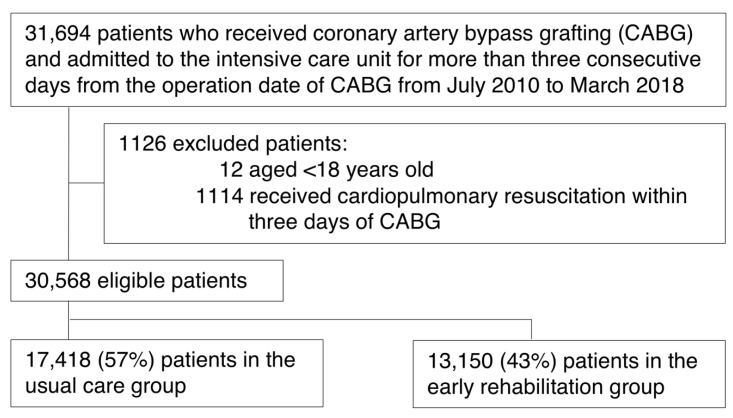
Patient inclusion/exclusion flowchart.

**Table 1 jcm-10-00618-t001:** Patients’ characteristics in the unweighted and weighted cohorts.

Characteristics	Unweighted Cohort			Weighted Cohort		
	Usual Care (*n* = 17,418)	Early Rehabilitation (*n* = 13,150)	ASD	Usual Care (*n* = 17,547)	Early Rehabilitation (*n* = 13,021)	ASD
Age, years, mean (SD)	71 (10)	71 (10)	4	71 (10)	71 (10)	0
Male, %	73	74	3	73	73	0
Current/past smoker, %	47	51	8	49	49	0
Body mass index, kg/m^2^, mean (SD)	24 (4)	24 (4)	4	24 (4)	24 (4)	0
Barthel Index score at admission, mean (SD)	80 (35)	88 (28)	23	83 (32)	83 (32)	2
Japan Coma Scale score at admission, %						
Alert	95	98	12	96	97	2
Dizziness	3	2	6	2	2	1
Somnolence	1	0	6	1	0	1
Coma	1	0	9	1	1	1
Calendar year, %						
2010–2011	21	8	37	16	15	2
2012–2013	30	21	21	26	25	1
2014–2015	26	31	11	28	29	1
2016–2017	23	40	37	30	31	2
Ambulance use, %	19	12	17	16	16	1
Emergency admission, %	32	23	19	28	28	1
Length of stay until surgery, %						
0 days	16	7	30	12	12	3
1–7 days	44	56	23	49	50	1
≥8 days	39	37	4	39	39	1
Annual hospital volume, per year, mean (SD)	23 (19)	23 (18)	3	23 (18)	22 (17)	1
Teaching hospital, %	74	67	15	70	70	1
Charlson comorbidity index score, mean (SD)	1.8 (1.3)	1.8 (1.3)	3	1.8 (1.3)	1.8 (1.3)	1
Comorbidities, %						
Chronic lung diseases	3	4	4	3	4	0
Cerebrovascular diseases	9	11	6	10	10	1
Peripheral vascular diseases	10	12	5	11	11	0
Diabetes mellitus	42	46	9	44	44	1
Hypertension	49	56	15	52	52	1
Chronic kidney diseases	20	18	6	19	19	0
Surgical characteristics						
Off-pump CABG, %	32	35	8	34	34	1
Diseased vessels ≥2, %	79	79	2	79	79	0
Concomitant valve replacement, %	28	27	2	28	28	0
Total anesthetic time, minutes, mean (SD)	526 (300)	489 (285)	13	511 (253)	510 (434)	0
Treatments within 2 days of CABG, %						
Invasive blood pressure monitoring	89	89	1	89	90	1
Central venous pressure monitoring	56	58	5	57	57	0
Pulmonary artery pressure monitoring	69	62	15	66	66	0
Supplemental oxygen	27	34	16	30	30	1
Mechanical ventilation	83	79	12	82	82	0
Renal replacement therapy	21	16	15	19	18	1
Mechanical circulatory support	27	14	33	21	20	3
Dopamine	76	71	11	74	74	0
Dobutamine	68	61	13	65	65	0
Noradrenaline	88	85	11	87	87	0
Adrenaline	17	13	12	15	14	2
Vasopressin	4	4	2	4	4	0
Beta blockers	60	64	8	61	61	0
Diuretics	69	72	6	70	70	0
Propofol	89	89	0	89	89	0
Midazolam	75	72	7	73	73	0
Dexmedetomidine	53	53	0	53	54	1
Antipsychotics	14	13	1	14	14	0
Stress ulcer prophylaxis	99	99	1	99	99	0
Enteral nutrition	4	4	2	4	4	0
Parenteral nutrition	1	1	3	1	1	0
Insulin	77	78	1	77	77	0
Red blood cells	85	78	17	82	81	1
Fresh frozen plasma	76	67	19	72	72	1
Platelets	52	40	24	47	47	1
Total fluids, L/day, mean (SD)	6.8 (2.8)	6.3 (2.3)	22	6.6 (2.7)	6.5 (2.5)	2

ASD, absolute standardized difference; CABG, coronary artery bypass grafting; SD, standard deviation.

**Table 2 jcm-10-00618-t002:** Outcomes in the unweighted and weighted cohorts using inverse probability of treatment weighting analyses.

Outcomes	Unweighted Cohort		Weighted Cohort			
	Usual Care (*n* = 17,418)	Early Rehabilitation (*n* = 13,150)	Usual Care (*n* = 17,547)	Early Rehabilitation (*n* = 13,021)	Differences (95% CI)	*p*-Value
Primary outcome						
Barthel Index score at discharge, mean (SD)	81.6 (34)	88.8 (27)	83.8 (32)	86.8 (29)	3.2 (1.5–4.8)	<0.001
Secondary outcomes						
In-hospital mortality, %	8.1	3.8	6.7	4.9	−1.8 (−2.6 to −1.0)	<0.001
Length of ICU stay, days, mean (SD)	8.2 (10)	7.0 (8)	7.9 (9)	7.4 (8)	−0.5 (−0.8 to −0.3)	<0.001
Length of hospital stay, days, mean (SD)	43.8 (44)	37.0 (29)	42.3 (43)	38.6 (31)	−3.7 (−5.2 to −2.2)	<0.001
Total hospitalization cost, ×10^5^ yen, mean (SD)	56.9 (34)	48.1 (23)	54.1 (31)	50.5 (24)	−3.6 (−5.1 to −2.1)	<0.001

CI, confidence interval; ICU, intensive care unit.

**Table 3 jcm-10-00618-t003:** Results of the complete case analyses.

Outcomes	Unweighted Cohort		Weighted Cohort			
	Usual Care (*n* = 12,908)	Early Rehabilitation (*n* = 10,442)	Usual Care (*n* = 17,547)	Early Rehabilitation (*n* = 13,021)	Differences (95% CI)	*p*-Value
Primary outcome						
Barthel Index score at discharge, mean (SD)	82.9 (33)	89.7 (25.8)	84.8 (31)	87.9 (28)	3.1 (1.2–5.0)	0.001
Secondary outcomes						
In-hospital mortality, %	7.6	3.6	6.3	4.6	−1.7 (−2.6 to −0.8)	<0.001
Length of ICU stay, days, mean (SD)	8.0 (9)	6.9 (8)	7.7 (8)	7.2 (9)	−0.5 (−0.8 to −0.2)	0.003
Length of hospital stay, days, mean (SD)	42.9 (44)	36.3 (29)	41.3 (43)	38.1 (31)	−3.2 (−4.8 to −1.6)	<0.001
Total hospitalization cost, ×10^5^ yen, mean (SD)	55.4 (32)	47.3 (22)	52.7 (30)	49.6 (24)	−3.1 (−4.6 to −1.6)	<0.001

## Data Availability

The datasets analyzed during the current study are not publicly available owing to contracts with the hospitals providing data to the database.

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
