# Peer review of "Effect of Early Rehabilitation on Physical Function in Patients Undergoing Coronary Artery Bypass Grafting: A Nationwide Inpatient Database Study"

_jcm, 2021, doi:10.3390/jcm10040618_

Round 1

Reviewer 1 Report

Ohbe et al. provide a database analysis including >30k patients after coronary artery bypass grafting. They analysed data using weighted propensity score models after multiple imputations of missing data and confirmed positive effects of cardiac rehabilitation phase I on activities of daily living, mortality, length of stay and hospitalization costs. Congratulation to the authors for this excellent work. The manuscript is well written and organized, comprehensive and transparent.

I have only one minor point: The conclusion of the safety of early rehabilitation is understandable but not clearly operationalized. You did not assess adverse events directly resulting from rehabilitation. I suggest adding an explanation (for example "... was safe as suggested by the low mortality ...").

In line 35, "for critically ill patients" seems dispensable to me since it is mentioned twice in this sentence.

Reviewer 2 Report

I read with interest the manuscript by Dr. Ohbe et al. This is a very important topic that is in great need of more investigation While the paper employs sophisticated methodology to account for its observational nature, there are still significant questions that remain unanswered. The primary question is that I am very skeptical that any degree of statistical manipulation can erase the fact that the Early Rehabilitation Group appears to be significantly less sick than the Usual Care Group. At the very least, I think a thorough review by a statistician is warranted.

Please ensure a native English speaker reviews the manuscript as there are grammatical issues including those of inadequate parallel structure and subject-verb mismatch.

Abstract

Please change “in-tensive” to “intensive” and “de-fined” to “defined”

Introduction

Please revise the first sentence to ensure there is parallelism (i.e do not provide a list of three and then say “which is termed” implying a singular object was preceding)

“ADL” should be changed to “ADLs” as is more frequently used in the literature

Methodology

While the statistical methodology certainly seems sophisticated, I believe that it could be more clearly explained as there are some puzzling data in Table 1.

First, why are there more patients in the WIEGHTED Usual Care cohort than in the Unweighted Cohort? Certainly, with propensity score matching that number should decrease. I simply do not understand how the weighted Usual care cohort can have 17.547 while the unweighted has 17,418.

Along a similar vein, I do not understand how the cohorts presented do not reflect the high number of sick patients.

For instance, in the supplementary data, Usual Care patients have nearly double the rate of mechanical circulatory support (27% vs. 14%), and higher rates of pulmonary arterial pressure monitoring, renal replacement therapy, ambulance use, dopamine, dobutamine, adrenaline, noradrenaline, mechanical ventilation, RBC transfusion, and FFP use. While no statistics are presented to determine if these differences are significant, I suspect many of them are. And, while some may be coincidental, all these taken together suggest a considerably sicker Usual Care group which would certainly explain why they were not offered early rehabilitation within the first 3 days after CABG. I recognize this was the purpose of the IPTW but the absolute standard differences for many of these marked differences is still greater than 1.

Results

Do the authors have any data on the amount (hours) of rehabilitation the Early Rehab group received compared to the Usual Care group? In-hospital mortality is not an appropriate surrogate for safety or efficacy, in my opinion. It is highly unlikely that several sessions of rehab in a CABG patient would provide a mortality benefit (rather, the mortality more realistically lies in the candidacy of that patient for rehabilitation).

What data do the authors present to substantiate the claim that “cardiac rehabilitation in the early phase 1 after CABG [is] safe?” (in conclusions). While this may be true, there are no data on adverse events from the rehabilitation intervention and thus I do not think it is appropriate to state this.

Overall, I think that review by a statistician is need to ensure that the methodology is valid. Perhaps out of ignorance, I am unconvinced by the conclusions made by the authors but believe there is potential. The results of this study, to me, simply suggest that there is a correlation between early rehabilitation and improved lengths of stay and survival given that absolute standardized differences are still >0 for many of the important variables indicating a sicker Usual Care population.

Round 2

Reviewer 2 Report

I appreciate the changes made by the authors and believe that the manuscript has been improved. The predominant remaining issue remains grammar and syntax.

The authors have still not explained why the Usual Care Weighted cohort has more patients than the Unweighted Usual Care cohort in Table 1. Were there crossovers?

Discussion – the last sentence of the first paragraph states “with lower in-hospital mortality AND total hospitalization costs, AND shorter length of ICU stay AND hospital stay.” This is awkward to read and all of these ‘and’s’ are not necessary. Please reword this as well as the description of table 2 which has a similar sentence.

Conclusion- the word “suggested” is used twice in one sentence – please reword
